# The Altruistic Behaviour of Consumers Who Prefer a Local Origin of Organic Food

**Adam Czudec** 

Institute of Economics and Finance, College of Social Sciences, University of Rzeszów, 35-310 Rzeszów, Poland; aczudec@ur.edu.pl

**Abstract:** Despite the fact that consumer behaviour in the organic foods market has been the subject of numerous studies in various countries around the world, little research has been devoted to the assessment of the importance of the altruistic behaviour of consumers who prefer a local origin of such food. Therefore, the aim of this paper was to determine the motives of organic food consumers for their interest in the local origin of food in the context of behaviour defined as either altruistic or egoistic. The study was carried out among 850 consumers of organic foods in Poland. The Kruskal–Wallis test and Dunn's post hoc test were used for the analysis of empirical data. This study shows that the emphasis on the importance of the local origin of this kind of food by organic food consumers is related to their awareness of the needs of other people; specifically, this is demonstrated by these consumers taking into account the importance of caring for the natural environment in their purchasing decisions. Therefore, this is an example of altruistic behaviour which also fits into the concept of reflexive localism. It was further determined that this consumer group has a stronger and more robust relationship with the organic food market than the market's other members.

**Keywords:** organic food; consumer behaviour; altruism; reflexive localism; local origin food

## 1. Introduction

The ever-progressing deterioration of the environment on a global scale, as a result of which existential risks are growing, has forced the need for changes in the field of natural resource management, including those used in food production. In the case of agriculture, the task of restricting the environmental impact caused by the common use of industrial production methods is gaining importance; this is because this practice results in not only low quality food but also in a decrease in biodiversity and in ongoing soil and water pollution [1,2]. Here, an alternative comprises the spreading of methods for agricultural raw materials production that do not further damage the condition of the environment and also contribute to its improvement [3]. Such a task is most completely implemented by organic holdings. The dynamics for their development largely depend on consumer behaviour, which creates the demand for certain foods and thus determines the direction of the actions of the agricultural sector in adapting to the needs of consumers.

The communication of the European Commission entitled the "New Green Deal" (communication from the commission to the European Parliament, the Council, the European Economic and Social Committee and the Regional Committee on an action plan for the development of organic production com/2021/141 final/2, https://eur-lex.europa.eu/legal-content/EN/TXT/?uri=CELEX%3A52021DC0141R%2801%29, accessed on 28 September 2021), assuming an increase in the contribution of arable land for organic crops from the current 8.5% to at least 25% by 2030, highlights, i.a., the importance of stimulating the demand for organic foods, increasing the trust in such food from consumers, and creating local organic food markets. Stimulating the demand and building trust will be implemented by means of educational and informational activities. On the other hand, the local context will be implemented by creating a new business model, referred to as the

'Bio-District', consisting of agreements between farmers, residents, and public authorities on the sustainable management of local resources. The aim of such projects is to enhance the use of the economic, social, and cultural potential of the local environment while maintaining and expanding local lifestyles, interpersonal relations, and the good condition of the environment.

Producers and consumers of organic foods view as particularly important the reduction in food transport costs by shortening supply chains and the consumer's appreciation of the value of local organic foods, which translates into higher market values. Research on the organic foods market conducted thus far has shown that its development is mostly determined by demand-shaping factors [4,5], among which the motives for purchasing organic food are the most important and determine consumer decisions. The most frequent motive mentioned by the majority of studies is the care for one's own health, independently of the consumer's level of income [6,7]. Taste and place of residence have equally high positions in the ranking the factors that determine organic food buying decisions [8]. Care for the condition of the environment and support for the local economy [9,10] and for animal welfare [11] are of lesser significance. The first of the aforementioned motives are characteristic of egoistic consumer attitudes, whereas the second group corresponds to altruistic traits [12–15], which are in turn linked to the moral values determining purchasing decisions [16]. Although care for the environment is an example of consumers for whom such a motive is important adhering to moral values, supporting the local economy through purchase decisions is assessed on a similar level only when consumers represent attitudes characteristic of reflexive localism. This means that a consumer's openness to foods produced in their region of residence is also an approval of the purchase of products present on the global market and which cannot be produced in the given local environment. In this case, the openness of consumers is exhibited by, i.a., the purchase of citrus fruit and other organic farming products originating from countries with lower economic development, which not only contributes to the dynamisation of economic and social growth in such states but also positively affects an improvement in the condition of the environment on a global scale. Such an attitude favours the implementation of sustainable development not only in the local environment, close to organic food consumers in wealthy societies, but also in countries with a lower level of economic development [17,18].

Thus, the reflexive localism approach may be an efficient method of caring for the environment, at the same time reducing the potential hazards resulting from the negative outcomes of globalisation on local economies and communities [19,20]. Contrary to reflexive localism (sometimes referred to as global reflexivity), defensive localism, which means closing to the international commercial exchange and a lack of approval for imported organic foods from developing countries, results in deepening economic and social inequalities, as well as reducing the possibilities of improving the condition of the environment on a global scale. In connection with the recent years' growing interest in local foods, as shown in the literature and among organic food consumers [21,22], the study objective was assumed as determining the motives for such an approach in the context of the different importance of altruistic and egoistic attitudes of organic food consumers preferring food of local origin and for the positive impact of organic agriculture on the global environment. The justification for such a direction of the study is the need to enrich the knowledge on local organic food consumer behaviour, which can be used for programming actions for increasing the demand for organic foods and promoting altruistic values among those consumers. It is one of the basic conditions for the dynamisation of the process of developing local organic food markets based on the concept of the common good, as well as a way towards sustainable consumption and promoting the attitude known as Ecological Citizenship [22,23]. It is also a method for enhancing the role of local organic foods markets in the global economy, eliminating the effect of substitution between organic and local food, which, as has been shown in the literature, is characteristic of some markets [24].

## 2. Literature Review and Research Hypotheses

Consumer market behaviour is motivated by different internal (resulting from the personality traits of each consumer) and external factors (created by the environment). These can be arranged in several groups: factors related to the traits of organic food (health and taste values, availability and convenience of shopping, high quality) [25,26]; personality traits of consumers (gender, age, education, place of residence) [27]; market-related factors (prices of organic foods, consumer income) [28]; socials norms (fashion, tradition, following the behaviour of others) [29]; environmental motives (natural environment protection, animal welfare, place of origin of organic food) [30,31].

As shown by the literature, none of the mentioned factors has the sole impact on consumer purchasing decisions, but not all of them act at the same time, and they do not have an equal effect on organic food purchasing motives.

Although the majority of studies show that the largest organic food consumer group mentions factors related to organic food characteristics in the first place among its purchasing motives, in the case of the other mentioned groups, their importance differs quite clearly between countries and regions [15,32,33]. Furthermore, some of them are decreasing in significance (social norms in highly developed countries), while others play an increasing role among organic food purchasing motives (the environmental motives). One such factor is its local origin, which, for consumers, does not only mean the shortening of supply chains, but the purchasing such foods is treated as a form of supporting local farmers and the local economy. Complex relationships between producers and consumers at a local level increase the amount of money in circulation, thus contributing to local development and enhancing local well-being and social value outcomes as they create and reproduce local interaction, social relationships, and civil society [34–36].

In this context, it appears significant to seek an answer to the question whether and to what degree organic food consumers are interested in its local origin. Studies conducted in Austria [37] and Sweden [38] have shown that organic food consumers exhibit a growing interest in its local origin, with the main causes typically being its high quality and support for the local economy. On the other hand, research conducted among organic food consumers in Denmark point to a low interest in its local origin [39]. In turn, a study from England has demonstrated that only a small portion of farmers running organic holdings was interested in placing their produce on local markets [40]. However, research from the USA [41] and Germany [42] has given basis to the conclusion that organic food consumers accepted the highest price for organic products that originate from local suppliers. No difference was observed between the price paid for local products without an organic certificate and organic products from outside of the local environment. All of this leads to the conclusion that the local origin of organic food may have a rather varied importance among consumers from different countries, and more detailed research on the causes and outcomes of such an approach is necessary [43]. In this case, particularly significant are the evaluations of the motives targeting 'localism', which, from the consumers' standpoint, has a geographic dimension and means the distance from the place of food production to the purchase location, typically not exceeding 80 to 100 km [44,45].

A characteristic of local food determining its popularity among consumers is mainly its freshness, but purchasing motives also include creating a close relationship with farmers and maintaining tradition in the local environment [46,47]. Another important motive is the support of local businesses and farmers [48] and care for the environment via shortening food supply chains and reducing transport costs, leading to a more sustainable development of agriculture [37,49].

Local food markets can also constitute a source of undesirable effects, which are manifested by consumer attitudes characterised by:

- Treating localism as a form of defence against the economic globalisation process, meaning closing to other markets [50];
- Particularity supporting local elites and leading to the exclusion of other food market members, eliminating the fair trade principle, and a deterioration of social justice,

which is exemplified by participation in campaigns organised under the 'Buy Local Food' headword [51,52].

Another risk appearing in the consumer environment is the a priori assumption that buying local food always contributes to supporting sustainable farming, which has been defined by Born and Purcell [53] as the 'local trap' because they determined that, in reality, local food production systems are generally no more sustainable than other systems.

Consumers who treat purchasing local food as a form of excluding the local environment from the global economy or viewing localism as an efficient method of isolation from other market participants exhibit an attitude defined as defensive localism [48,50].

In order to avoid the negative consequences of the 'local attitude' to the production and distribution of food, it is necessary that different institutions promote a policy that favours attitudes among consumers which are characterised by interest in purchasing local food but devoid of particularity and understanding the important role of social justice, not only in a local but also a global dimension. This understanding of localism may be an important factor improving the competitiveness of local environments and regions and thus fitting in with global circulation. Such attitudes are defined as 'reflexive localism' [17,20,50,54].

Considering that the research conducted thus far does not provide a clear answer to the question of the relationship between motives for purchasing organic and local food, including the interest of consumers of organic food in its local origin, a more detailed study in this field appears justified. By this in particular the authors mean the assessment of the behaviour of consumers, who, by buying organic food, notice the significance of its local origin. It can be assumed that consumers highlighting this motive for organic food purchase may bring a positive outcome in the form of greater interest in the local market from farmers running organic holdings [40], which may prevent the appearance of negative outcomes of organic food market becoming similar to that of the global market [55]. Furthermore, the local origin of organic food may ensure higher income for its producers [41,42].

Independently of whether consumers are interested in purchasing organic food value its local origin, or whether it is a trait that does not affect purchasing decisions, consumers' behaviour may be dominated by an egoistic or altruistic approach [56]. The former means prioritising personal motives (care for one's health; high quality of food; better taste), whereas the altruistic approach is manifested by consumers taking into account such motives as care for the environment and the protection and animal welfare [33] or supporting local products and, in a broader meaning, the local economy [12,57,58].

Studies conducted thus far on this aspect of consumer behaviour have focused on either local food or (in the case of organic food) they did not include the division of consumers into those who prefer a local origin of organic food and others [14,16,59]. Therefore, this study aims at filling this gap (at least partially) by means of verifying the following hypothesis:

**Hypothesis 1 (H1):** *Consumers who prefer organic food produced in their region of residence (local) present altruistic attitudes to a greater degree than other consumer groups.*

Considering the results of studies on local food [12] and organic food consumer behaviour [58], motivated by altruistic premises showing that this group of consumers more commonly accepts the higher price of organic food and buys it more frequently, the following hypothesis was assumed for verification:

**Hypothesis 2 (H2):** *Consumers who prefer organic food originating from their region of residence have stronger ties to the market of such food than other consumer groups.*

## 3. Materials and Methods

The analysis leading to the realisation of the study objective and verification of the hypotheses was conducted on the basis of a survey among 850 consumers of organic food in

Poland. The research sample was randomly selected on the basis of age (18 years and more), gender, education level (primary; secondary; higher), the number of persons in the family, form of professional activity (wage work; self-employed; pensioner; student; unemployed), and place of residence of the respondents (rural areas; towns with different number of inhabitants). The study was carried out in December 2020 by a specialised research agency using the CAWI method among those consumers who made purchases at least once a month. Following the assumptions of the study and the analysis leading to verification of the hypotheses, there are three variables characterising organic food consumer behaviour: preferences as to the place of organic food production; altruistic attitudes in consumer behaviour; and the strength of the relationship between consumers and the organic food market.

Within the first variable, consumers were divided into three groups: preferring food in their region of residence; preferring food produced in Poland, regardless of the region; and other (preferring food produced abroad or those who do not pay attention to the place of production).

Subsequently, the intensity of altruistic attitudes and the ties of consumers to the organic food market were assessed for each group.

Altruistic attitudes were identified based on four empirical indices related to organic food purchasing motives: purchases motivated by supporting the local economy; taking into account care for the environment in purchasing decisions; consumers making purchasing decisions on the basis of the place of origin of raw materials used to produce organic food; consumers taking into account the distance over which food is transported from the producer to the place of sale.

Similar criteria for altruistic attitude identification (also known as ethical values) among organic food consumers were used in the study of other authors [30,60–64].

In the survey, consumers exhibited five main motives for purchasing organic food, at the same time arranging them in a hierarchy. Thus, in order to determine the level of intensity of altruistic attitudes for each of the three consumer groups, each of the selected motives were assigned a point score depending on the place of such motive in the ranking (five points for the most important motive, four for the second most important, and one for the last one of the five selected by consumers). The higher the total number of points, the higher the intensity of altruistic attitude occurrence among consumers.

In turn, to determine the strength of the relationship between consumers and the organic food market, three measures were used:

- Frequency of purchasing organic food (daily—4 points; several times a week—3; once a week—2; once a month—1);
- Time over which consumers have been purchasing organic food (for several years—4 points; for 1 year—3; for several months—2; for several weeks—1);
- Monthly expenditure on organic food (over PZ 500 (Polish national currency—PZ 1 = approx. EUR 0.22)—4 points; PZ 200–500—3; PZ 100–199—2; under PZ 100—1 point).

Similarly, as in the case of the measures of altruism, a higher number of points was treated as a proof of stronger ties of consumers to the organic food market.

In order to test the hypotheses, measures of altruism and connections with the market were juxtaposed with the qualitative variable describing consumer preference for the place of origin of organic food. Therefore, in order to select appropriate statistical methods, a normal distribution of variables was verified including division into consumer categories (Table 1).

**Table 1.** Normal distribution analysis—Shapiro–Wilk test results.

| Group | Altruism Level Scale | Scale of the Level of Connection with the Organic Food Market |
|---|---|---|
| preferring food produced in their region of residence | W = 0.98034, *p* = 0.00008 | W = 0.96281, *p* = 0.00000 |
| preferring food produced in Poland | W = 0.98105, *p* = 0.00057 | W = 0.96220, *p* = 0.00000 |
| other | W = 0.96764, *p* = 0.00021 | W = 0.96113, *p* = 0.00004 |
| Total | W = 0.98231, *p* = 0.00000 | W = 0.96869, *p* = 0.00000 |

Source: own study.

Both in the individual groups as well as within all of the respondents, the conducted Shapiro–Wilk tests did not reveal variable distributions close to normal distribution. Therefore, in order to test the hypotheses, the Kruskal–Wallis non-parametric variance analysis of ranks (Kruskal–Wallis test) was used, followed by post-hoc testing with Dunn's test.

## 4. Results

Women were predominant among the respondents (58%), while people with higher education constituted 48% of the total number of consumers. Approximately 38% of the respondents live in the countryside, while 33% live in cities above 100 thousand citizens. Of the total number of respondents, employed persons were predominant (65%), and only 10% were self-employed. The remaining respondent groups are as follows: pensioners (8%), students (9%), and the unemployed (8%). In terms of the level of consumer wealth measured by income per one family member, 31% of the total group had a relatively high monthly income (over PZ 2500), 39% had average income (PZ 1500–2500), while other consumers belonged to the less wealthy group (income per one family member did not exceed PZ 1500). Like most of the above-mentioned characteristics, the number of people in the families of the respondents was quite typical for Polish society, that is, 53% of the respondents lived in families of 3–4 people, while 7% were single people, and 8% were members of families with at least 6 people. The mean age of respondents is 38 years, with a rather high age diversity (standard deviation—14).

To verify the Hypothesis (H1) assuming that consumers who prefer organic food produced in the region of their residence (to a greater extent than others) present altruistic attitudes, the results illustrating the level of altruism were compared with the preferences regarding the place of production of organic food (Table 2).

**Table 2.** Altruism level among consumers vs. their preferred place of organic food production.

| Preferred Place of Food Production | *n* | Altruism Level | | | |
|---|---|---|---|---|---|
| | | Mean | Median | Total Ranks | Mean Rank |
| in the region of residence | 361 | 9.47 | 9.0 | 166,257.5 | 460.6 |
| in Poland regardless of region | 297 | 8.94 | 8.0 | 125,041.5 | 421.0 |
| other | 192 | 8.28 | 8.0 | 70,376.0 | 366.5 |
| Statistical significance: | | H = 18.69684, *p* = 0.0001 | | | |

Source: own study.

Based on the calculations in Table 2, it can be concluded that statistically significant differences exist between the selected groups of consumers in terms of the intensity of features characterising altruistic attitudes, as evidenced by the Kruskal–Wallis test.

In order to provide greater detail of the results, a post-hoc Dunn's test was performed (Table 3).

**Table 3.** Altruism level—Dunn's test results.

| Preferred Place of Food Production | in the Region of Residence | in Poland Regardless of Region | Other |
|---|---|---|---|
| in the region of residence | - | 0.119539 | 0.000054 |
| in Poland regardless of region | 0.119539 | - | 0.049733 |
| other | 0.000054 | 0.049733 | - |

Source: own study.

On this basis, it was found that consumers preferring foreign food or not paying attention to the place of its production were characterised by a significantly lower level of altruism ($p < 0.05$) than the other two groups of consumers. On the other hand, no statistically significant differences were found in terms of the intensity of altruistic values between consumers who prefer organic food produced in the region of their residence and those who prefer food from Poland, regardless of the region, which does not allow for the assumption of the H1 hypothesis. An interpretation of this result can be that consumers interested in the local and national origin of organic food are similarly motivated by altruistic values when making purchasing decisions. On the other hand, other consumers attach more importance (compared to other motives for purchase) to the health and taste of organic food, while the problem of caring for the environment or the motive of supporting the local economy is of less importance to them.

Studying the significance of altruistic attitudes among organic food consumers would have little cognitive significance (particularly from an economic point of view) if it was not related to the assessment of the strength of consumers' ties with the market for such food. In this study, it was assumed that consumers who prefer food produced in the region of their residence not only display altruistic attitudes but are more closely related to the market for such food, and the strength of this relationship is measured by the frequency of purchases, the period of presence on the organic food market and the amount of money spent on purchases of such food.

The Kruskal–Wallis test results demonstrated statistically significant ($p < 0.05$) differences in terms of ties to the organic food market between the distinguished consumer groups (Table 4).

**Table 4.** Level of relationship of consumers with organic food market.

| Preferred Place of Food Production | n | Measures of the Level of Connection with Organic Food Market | | | |
|---|---|---|---|---|---|
| | | Mean | Median | Total Ranks | Mean Rank |
| in the region of residence | 361 | 8.20 | 8.0 | 174,519.0 | 483.4 |
| in Poland regardless of region | 297 | 7.88 | 8.0 | 129,365.5 | 435.6 |
| other | 192 | 6.93 | 7.0 | 57,790.5 | 301.0 |
| Statistical significance: | | H = 72.36023, $p = 0.0001$ | | | |

Source: own study.

This means that the organic food consumers covered by this study differ not only in terms of preferences for the place of origin of such food but also in the strength of their ties to such a market. In order to provide a more precise determination of the level of diversity of these connections, a post-hoc Dunn's test was carried out (Table 5).

**Table 5.** Level of relationship with market—Dunn's test results.

| Preferred Place of Food Production | in the Region of Residence | in Poland Regardless of Region | Other |
|---|---|---|---|
| in the region of residence | - | 0.038514 | 0.000000 |
| in Poland regardless of region | 0.038514 | - | 0.000000 |
| other | 0.000000 | 0.000000 | - |

Source: own study.

Additional analysis showed that statistically significant ($p < 0.05$) differences occurred between all selected combinations (pairs) in terms of the strength of ties to the organic food market. The strongest relationship with the market characterised those who preferred food produced in the region of their residence, while the smallest were those who did not pay attention to the place of production or prefer food from abroad. Therefore, the results presented herein allow for the assumption of the H2 hypothesis.

**5. Discussion**

All of this may mean that consumers who prefer organic food from their region of residence are not only more motivated by altruistic values than others but also have a stronger relationship with the market for such food. It should be also emphasised that this consumer group has a relatively high contribution in the total number of people buying organic food in Poland (in this study, this amounts to about 42%). However, it is not as high a percentage as, for instance, in Romania, where 65% of consumers prefer a local origin of organic food [31]. The important role of the strength of consumer relations with the organic food market is demonstrated by research conducted in Canada [58] and China [33] showing that consumers who buy organic food frequently (at least once a week) more often mention the need to support the local economy as a motive for purchase, and treat shopping as a form of care for the environment compared to those consumers who buy such food rarely [65]. In this aspect, these findings are consistent with the results presented here. On the other hand, Thomas & Gunden [66] arrived at different conclusions because, based on research conducted among US consumers, they concluded that organic food was perceived as a way to meet the health needs of consumers, but without any positive impact on the natural environment and the local economy. However, in Indonesia, both altruistic and egoistic motives have a strong influence on the behaviour of organic food consumers [15].

Based on the research results presented in this study, it can be concluded that the behaviour of consumers who prefer organic food produced in their region of residence corresponds more closely to the features of reflexive localism than in the case of other consumers. This is evidenced by a relatively high index illustrating the level of altruism in this group of consumers, the components of which, apart from supporting the local economy, include such motives as care for the environment by producers of such food or taking into account the environmental hazards resulting from the costs of transporting raw materials and ready-made organic products. The interest in the condition of the natural environment when making organic food purchasing decisions is treated as evidence of care for other people, which is the essence of altruistic behaviour [13,14,57]. All of this means that preferring a local origin of organic food by the surveyed consumers shall be viewed as an example of pro-social behaviour, taking into account the welfare of others. In this aspect, our results differ from the study of Birch et al. [56], conducted among consumers of local food (but not organic), showing that egoistic attitudes (health orientation; safety of food) are dominant over altruistic behaviour. This can be treated as evidence confirming the statement on the advantage of reflexive localism attitudes among the surveyed consumers of organic food who prefer its local origin. At the same time, it can be concluded that for a large group of the surveyed consumers, organic food is not a substitute for local food (and

vice versa), while its local origin is treated here as an important motive for its purchase, increasing the relationship between consumers and the market.

## 6. Conclusions

The organic food market is developing primarily due to the growing demand for such food. In turn, the demand is formed by different factors, among which the attitude of consumers to organic food may be of key significance. Not only can it be seen as a way to eat properly and take care of one's own health, but buying organic food can also be treated as a form of activity for the welfare of others by supporting the local economy or caring for the environment. As the research presented here shows, the latter approach (typical of altruistic attitudes) is quite apparent among organic food consumers in Poland, especially in the group which prefers organic food produced in their region of residence. In this case, emphasising the important role of the local origin of such food does not mean being closed to the external environment, but it is related to the perception of the needs of other people by taking into account the important role of care for the environment in purchasing decisions. Thus, such attitudes are referred to as reflexive localism.

An important conclusion drawn from this study appears to be the statement of the more profound relationship with the organic food market exhibited by those consumers who prefer its local origin. The dissemination of such attitudes could therefore contribute to building strong and lasting relationships between farmers, processors, and consumers of organic food at the local level, which in turn could prevent the negative consequences of the organic food market becoming similar to the global market, as highlighted in the literature [29–31,34], and would also contribute to strengthening the development of peripheral rural areas.

## 7. Limitations and Implications for Research

The research results presented in this paper concern the problem of altruistic behaviour of organic food consumers preferring its local origins, which has been poorly recognised, not just in Poland, and indicate the need for a more detailed study of this issue in various places around the world [67,68]. Although the study has been based on a large population of respondents, it cannot be treated as fully representative of all consumers of organic food in Poland, which shall be treated as its limitation. Furthermore, the study does not provide an answer to the question about the types of altruism mentioned in literature and present in the behaviour of organic food consumers [13]. More detailed research is also needed on the importance and effects of 'localism' as a motive for purchasing organic food, not only from the point of view of consumer behaviour but also the expectations of organic farmers, especially since, according to the literature, farmers in some countries have little interest in the development of local organic food markets [40]. Therefore, this can be a factor limiting the possibilities for the development of local markets of such foods.

## 8. Implications for Practice and Society

The dissemination of altruistic attitudes among organic food consumers preferring its local origin is important in every society. As it results from the research presented here, such a group of consumers in Poland has stronger ties with the organic food market compared to those consumers for whom the local origin of such food is of no significant importance. Moreover, for a large group of consumers, buying local organic food is also motivated by supporting the development of the local economy and actions to improve the condition of the natural environment. However, the scale of popularisation of altruistic attitudes among organic food consumers will depend on the effective implementation of educational programmes among the general population aimed at expanding knowledge about the positive effects of the development of local organic food markets. Institutions responsible for sustainable development at the local, national, and global levels have an important role to play in this aspect because their task should be popularising such a consumption model, which does not only ensure food safety for each resident but also

contributes to the improved quality of life of people on a worldwide scale by means of caring for the environment.

**Funding:** This research received no external funding.

**Institutional Review Board Statement:** Not applicable.

**Informed Consent Statement:** Not applicable.

**Data Availability Statement:** Not applicable.

**Conflicts of Interest:** The author declares no conflict of interest.

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
