# Peer review of "The Altruistic Behaviour of Consumers Who Prefer a Local Origin of Organic Food"

_agriculture, doi:10.3390/agriculture12040567_

Round 1

Reviewer 1 Report

The article addresses a relevant topic. However, the research question, as well as the general objective of the study, do not seem to me to address theoretical and/or practical gaps. Consumer behavior is a widely debated subject in the literature. This has been happening for decades. There are consolidated theories about influencing factors, types of decision-making processes, motivations, values and attitudes. Even with regard to pro-environmental consumption behavior, much progress has already been made in the theoretical and practical field. In this sense, the article fails in its introduction, literature review and discussion of the results. Despite a considerable sample, there is nothing new in the survey responses. I suggest a quick search on Scopus on the topics of pro-environmental behavior and consumption of organic food, so that you can confirm what I'm saying.

Authors must:
- qualify its introduction, bringing the theory consolidated on the subject, and justifying, perhaps by the country of application of the study, its importance;
- qualify the literature review (I suggest you take advantage of similar studies already published and enrich your reasoning... it is very fragile compared to everything we have available on consumer behavior, sustainable consumer behavior and purchase of organic products).
- considering what you expanded and qualified in the introduction and in the review, it will be possible for you to establish a more robust discussion regarding the advance that this study presents. If not for theory (and it seems to me that not), but for practice and society.

Author Response

Dear Reviewer ,

Thank you for preparing a detailed review of my article and for formulating comments improving its quality . I tried to follow most of the review suggestions, prepared by each of the three reviewers. I hope that the revised version of the article will meet the editors' expectations and will be published in " Agriculture ".

Best regards,

Adam Czudec

Reviewer 2 Report

Thank you for giving me this opportunity to review this manuscript. This manuscript is about the The altruistic behaviour of consumers who prefer a local origin of organic food. I have some suggestions for improving this manuscript for readers.

1. Introduction

The first and second paragraphs (line 21-53) in introduction should be supported by previous research or reliable information.

2. Introduction

If the authors want to use a concept or a guiding theory like the reflective localism, the defensive localism, and Ecological citizenship, these terms should be clearly defined. These terms seem to be very important, but as a reader, I somewhat feel the terms are suddenly shown without enough explanation.

3. Introduction

Some readers may not understand the distinguishing features of organic foods and consumers between the developed and the developing countries. The authors should provide sufficient explanations for enhancing the authors’ arguments in introduction.

4. Research purpose

At the end of the introduction, the purpose of this study needs to be provided clearly. The clear aims and objectives of this research will guide the readers for understanding this manuscript.

5. Literature review

The bullet points in the first paragraph would be not appropriate in a research paper. Please use a table if the author want to provide a brief content of previous literature or write full sentences.

6. Literature review

The author may need to reorganize the contents of literature review and provide supports for justifying the variables in the hypotheses. All concepts and important determinants should be defined and justified.

7. Materials and methods

As I suggested in literature review, the bullet points in materials and methods may not be appropriate. Please try to write full sentences for readers.

8. Methods

The data collection, sampling techniques, measurements, and analysis procedures should be more clearly provided from Lines 204 to 263.

hods

The analysis results are somewhat descriptive. Is there any better solution to show the different features of organic food consumers?

10. Results

From Lines 377, If the authors did not ask about the COVID-19 pandemic in the measurement, the assumptions in the results would be not recommended.

11. Conclusions

The contributions should be highlighted in discussions and conclusions.

I recommended to provide theoretical and practical implications separately in discussions and conclusions.

Author Response

(The authors gave the same response as above.)

Reviewer 3 Report

Dear authors,

Thank you for the possibility to read and evaluate your paper.
I am sending you my feedback in terms of Originality, Relationship to Literature, Methodology, Results, and Implications for research, practice and/or society.
My overall evaluation of the paper is a major revision.

1.    Originality
The theme of the article is actual and it brings new information on the altruistic behaviour of consumers on the local origin food market. The abstract should be more clear what is the aim of the article and what are the main findings. The aim of the article is quite well developed and corresponds with the article title. 

2.    Relationship to Literature
The relationship to literature is also well developer and consists of actual papers from the last five years

3.    Methods
The Methods part is quite clear. The sample selection should be presented in more detail as well as the demographic structure of the entire sample. Additionally, I miss some research models that would describe relations between hypotheses. I also miss a list of all items included in the questionnaire. Therefore, I cannot evaluate what questions from the questionnaire are connected with hypotheses.

4.    Results
The Results part is rather weak because the statistical methods used are basic (just statistical tests) and perhaps logistic regression could be used to explore relations among variables. The discussion part is completely missing and shud discuss the results of the study compared to the previous studies (mentioned in the Literature Review section).

5.    Implications for research, practice and/or society
Implications for research, practice and/or society are completely missing.

Author Response

(The authors gave the same response as above.)

Round 2

Reviewer 1 Report

I would like the authors to pay attention to the considerations made in the first review, and to send a detailed response letter, item by item, stating what was done and what was not done. Both situations must be justified.

Author Response

Dear Reviewer,

Thank you for preparing a second round review of my paper. I’ve read very carefully all the suggestions to improve the quality of the article, mainly related to the Introduction and Literature Review. I tried to enrich this part of the article with the latest literature on organic consumers altruistic behavior in relation to the local origin of organic food. I’ve used in the Literature Review all the works on this subject that I managed to find. So it is difficult for me to agree with your opinion that there is already extensive literature on this aspect of organic consumers behavior, which is the main subject of my research. Moreover, it seems to me that I have explained in sufficient detail (using the literature) the characteristics of the altruistic behavior of organic food consumers. By the way, I don't really understand the reasons why in the first review Research Design and Methods where appropriate and adequately described (checked: yes), while the next review- in your opinion-  this part of the article  “can be improved”. I hope that this explanations, as well as the changes introduced to the article after the reviews will meet your expectations and will allow it to be published in “Agriculture”.

Best regards,

Adam Czudec

Reviewer 2 Report

Overall, more citations for supporting authors’ argument are added in the manuscript. 

In method, I suggest that all contents should be writtin in full sentences rather than in bullet points. Some are revised but I can still see bullet points in materials and methods. 

1. introduction 

I am not sure why the upper letters and URL are important and are presented in introduction. 

2. introduction 

one sentence is a paragraph. The manuscript should be overally revised for increasing readerability. 

3. Overall, I found some gramatical errors in revised sentences. Proofreading should be done before submitting the manuscript. 

Author Response

Dear Reviewer,

Thank you for preparing a second round review of my paper. I tried as much as possible to take into account your comments, which improve the quality of the article. I understand your suggestion for a broader explanation of "Reflexive localism", "Defensive localism" and "Ecological citizenship". It seems to me that their meaning has been clarified in the literature review with appropriate citations, which allow any interested reader to get to know they full content, taking into account different contexts. As you suggested in review, I decided to remove the bullet points from my paper, as well as the sentence about Covid-19, as the assumption in the Results. I hope that this explanations, as well as the changes introduced to the article after the reviews will meet your expectations and will allow it to be published in “Agriculture”.

Best regards,

Adam Czudec

Reviewer 3 Report

Dear authors,

Thank you for the possibility to read and evaluate your paper.
I am sending you my feedback in terms of Originality, Relationship to Literature, Methodology, Results, and Implications for research, practice and/or society.
My overall evaluation of the paper is a minor revision.

1.    Relationship to Literature
The relationship to literature should consist of actual papers from the last five years. Add them to the manuscript, please.

2.    Methods
The sample selection must be presented in more detail as well as the demographic structure of the entire sample. Additionally, I still miss some research models that would describe relations between hypotheses. I also miss a list of all items included in the questionnaire. Therefore, I still cannot evaluate what questions from the questionnaire are connected with hypotheses.

3.    Results
The Results part is rather weak because the statistical methods used are basic (just statistical tests) and perhaps logistic regression could be used to explore relations among variables. The discussion part is completely missing and shud discuss the results of the study compared to the previous studies (mentioned in the Literature Review section). Definitely, I recommend dividing the Results and Discussion into two separate parts and improving the Discussion part significantly.

4.    Implications for research, practice and/or society
Implications for research, practice and/or society are completely missing.

Author Response

Dear Reviewer,

Thank you for preparing a second round review of my paper. I tried as much as possible to take into account your comments, which improve the quality of the article. I hope, you will find in the paper, according to your suggestion, more detailed characteristics of demographic structures of the sample, dividing the Results and Discussion into two separate parts, or improved Conclusions and Implications for practice and society. I tried to make more clear the Abstract as well. Unfortunately, some of the statements from the second round of review are incomprehensible to me. I do not know why in the second review in the section on Relationship to Literature there is the sentence “the relationship to literature should consist of actual papers from the last five years. Add them to the manuscript”. However, in the first review "the relationship to the literature is well developed and consists of actual papers from the last five years”. I also can’t agree which your opinion that “the discussion part is completely missing and should discuss the results of the study compared to the previous studies”. I would like to point out that in this part of the article there are comparisons with the results of similar studies in Romania, Canada, the USA and other countries. You also suggest application of logistic regression to explore relations among variables. However, this method could be useful when the dependent variable is dichotomous. In the case of my research this condition is not met. I hope that this explanations, as well as the changes introduced to the article after the reviews will meet your expectations and will allow it to be published in “Agriculture”.

Best regards,

Adam Czudec